# Improving uptake of prevention of mother-to-child HIV transmission services in Benue State, Nigeria through a faith-based congregational strategy

Michele Montandon[1]*, Timothy Efuntoye[2], Ijeoma U. Itanyi[3,4], Chima A. Onoka[3,4], Chukwudi Onwuchekwa[5], Jerry Gwamna[2], Amee Schwitters[2], Chibuzor Onyenuobi[2], Amaka G. Ogidi[3], Mahesh Swaminathan[2], John Okpanachi Oko[5], Gbenga Ijaodola[6], Deborah Odoh[6], Echezona E. Ezeanolue[3,7,8]

1 Division of Global HIV and TB, Centers for Disease Control and Prevention, Atlanta, GA, United States of America, 2 Centers for Disease Control and Prevention, Abuja, Federal Capital Territory, Nigeria, 3 University of Nigeria Center for Translation and Implementation Research, Nsukka, Enugu, Nigeria, 4 Department of Community Medicine, University of Nigeria, Nsukka, Enugu, Nigeria, 5 Caritas Nigeria, Abuja, Federal Capital Territory, Nigeria, 6 Nigeria Federal Ministry of Health, National AIDS and STI Control Program, Abuja, Federal Capital Territory, Nigeria, 7 Healthy Sunrise Foundation, Las Vegas, NV, United States of America, 8 Department of Pediatrics and Child Health, College of Medicine, University of Nigeria, Nsukka, Enugu, Nigeria

* nkf3@cdc.gov

Data Availability Statement: All relevant data are within the manuscript and its Supporting Information files.

## Abstract

### Background

Nigeria has low antiretroviral therapy (ART) coverage among HIV-positive pregnant women. In a previous cluster-randomized trial in Nigeria, Baby Shower events resulted in higher HIV testing coverage and linkage of pregnant women to ART; here, we assess outcomes of Baby Shower events in a non-research setting.

### Methods

Baby Shower events, including a prayer ceremony, group education, music, gifting of a "mama pack" with safe delivery supplies, and HIV testing with ART linkage support for HIV-positive pregnant women, were conducted in eighty sites in Benue State, Nigeria. Client questionnaires (including demographics, ANC attendance, and HIV testing history), HIV test results, and reported linkage to ART were analyzed. Descriptive data on HIV testing and ART linkage data for facility-based care at ANC clinics in Benue State were also analyzed for comparison.

### Results

Between July 2016 and October 2017, 10,056 pregnant women and 6,187 male partners participated in Baby Shower events; 61.5% of women attended with a male partner. Nearly half of female participants (n = 4515, 44.9%) were not enrolled in ANC for the current pregnancy, and 22.3% (n = 2,241) of female and 24.8% (n = 1,532) of male participants reported

**Funding:** This research has been supported by the President's Emergency Plan for AIDS Relief (PEPFAR) through the Centers for Disease Control and Prevention (CDC) under the terms of grant number 5NU2GGH000934. PEPFAR/CDC funded Catholic Caritas Foundation of Nigeria (CCFN), Author JOO, as the implementing partner for this project. CDC Atlanta and CDC Nigeria co-authors provided technical support and participated in study design, data analysis, decision to publish, and preparation of the manuscript. The Fogarty International Center of the US National Institutes of Health (NIH) under grant no. R21TW010252; supported author EEE (URL: https://projectreporter.nih.gov/reporter_SearchResults.cfm?icde=52475683). The funder has no role in study design, data collection and analysis, decision to publish, or preparation of the manuscript. Eunice Kennedy Shriver National Institute of Child Health & Human Development of the NIH under grant nos. R01HD087994 and R01HD087994-S1; supported authors EEE and CAO (URL: https://projectreporter.nih.gov/reporter_SearchResults.cfm?icde=52475683). The funder has no role in study design, data collection and analysis, decision to publish, or preparation of the manuscript.

**Competing interests:** No authors have competing interests.

they had never been tested for HIV. Over 99% (n = 16,240) of participants had their HIV status ascertained, with 7.2% of females (n = 724) and 4.0% of males (n = 249) testing HIV-positive, and 2.9% of females (n = 274) and 2.3% of males (n = 138) receiving new HIV-positive diagnoses. The majority of HIV-positive pregnant women (93.0%, 673/724) were linked to ART. By comparison, at health facilities in Benue State during a similar time period, 99.7% of pregnant women had HIV status ascertained, 8.4% had a HIV-positive status, 2.1% were newly diagnosed HIV-positive, and 100% were linked to ART.

## Conclusion

Community-based programs such as the faith-based Baby Shower intervention complement facility-based approaches and can reach individuals who would not otherwise access facility-based care. Future Baby Showers implementation should incorporate enhanced support for ART linkage and retention to maximize the impact of this intervention on vertical HIV transmission.

## Introduction

Nigeria, the most populous country in Africa with an estimated population of over 200 million, accounted for nearly fifteen percent of all newly acquired global HIV infections in children in 2019, with 22,000 estimated new child infections [1]. Despite the presence of a prevention of mother-to-child HIV transmission (PMTCT) program including HIV testing and antiretroviral therapy (ART), uptake of PMTCT services in Nigeria is low, with 43% of pregnant women living with HIV on ART and a mother-to-child transmission (MTCT) rate of 22% [1]. This low ART coverage among pregnant women can be partially attributed to low facility attendance for antenatal care (ANC) in pregnancy and low rates of health facility deliveries, since facility-based care provides an opportunity for HIV testing and enrollment in PMTCT services. In the 2018 Nigeria Demographic and Health Survey (NDHS), 67% of women who gave birth in the five years preceding the survey received ANC from a skilled provider, and 57% had at least four ANC visits. Only 39% of women delivered their last live birth in a health facility, with large variations in facility delivery by zone and state within the country [2]. In the 2018 Nigeria HIV/AIDS Indicator and Impact Survey, 77% of women who gave birth in the previous three years reported attending ANC, and 40% reported knowing their HIV status during pregnancy [3].

Various studies have demonstrated the effectiveness of engaging religious and community leaders to improve ANC attendance, HIV testing among pregnant women, and other PMTCT outcomes [4–6], and how faith-based responses to HIV differ from and may complement secular responses [7]. In the 2018 NDHS, 46.0% of Nigerians identified their religion as Catholic or other Christian, 53.5% Islam, and 0.5% Traditionalist or other [2]. Nigeria has a strong network of faith-based institutions with high religious service attendance, and faith plays a strong role in the social life of Nigerians [8].

In 2013, based on the strength of religious institutions and leaders in Nigeria and evidence supporting community engagement approaches for PMTCT, a cluster-randomized trial, the Healthy Beginning Initiative (HBI), tested the effectiveness of a faith-based congregational strategy for HIV testing among pregnant women in Enugu State, Nigeria [9]. The approach modified the baby shower that commonly occurs in the community (a reception held in

honour of a pregnant woman where she plays pregnancy-related games and receives gifts from friends) into a celebratory gathering held at the church, with prayer, singing, dancing, group education, and health screening, including HIV testing for pregnant women and their male partners [10]. The HBI trial demonstrated improved uptake of HIV testing during pregnancy (OR 11.2, 95% CI 8.77–14.25), linkage to care before delivery (OR 6.2, 95% CI 2.14–18.25), and male partner testing (OR 12.0, 95% CI 9.63–14.79) in the intervention group compared to the controls [11].

Based on the favorable results of the HBI trial and persistent gaps in PMTCT service uptake, the US Centers for Disease Control (CDC)-Nigeria and the US National Institutes of Health supported Catholic Caritas Foundation of Nigeria (CCFN), APIN, University of Nigeria and the HBI research team to implement Baby Showers in Benue State, Nigeria from July 2016 to December 2018. In this paper, we report the findings from Baby Shower events conducted between July 2016 to October 2017, the time period for which linkage to ART was systematically collected for HIV-positive pregnant women and referred to as the study period. We also compare HIV testing coverage, positivity, yield, and ART linkage from Baby Shower events to the standard approach of facility-based ANC testing for pregnant women in Benue State in order to determine how Baby Shower results in the community setting compare to the standard of care.

## Materials and methods

### Implementation setting and site selection

Benue State is located in north-central Nigeria, is predominantly Christian, and is one of the states with the highest HIV prevalence in Nigeria (4.8% HIV prevalence among age 15–64 year-olds in Benue State vs. 1.4% nationally) [3]. Within Benue State, health facilities with HIV and PMTCT programs receive support through the US President's Emergency Plan for AIDS Relief (PEPFAR), and data on HIV testing among pregnant women, ART uptake, and other HIV indicators are reported by health facilities to PEPFAR on a quarterly basis. In ANC settings, all pregnant women with unknown HIV status should be tested for HIV at the first ANC visit per national guidelines [12].

Across 12 local government areas (LGAs) in Benue State, 101 churches were assessed and, of these, eighty churches were selected and enrolled into the Baby Showers program. Sites were selected based on the capacity and willingness of the church, congregational size, and accessibility, including proximity to a health facility.

### Description of Baby Showers implementation

A detailed description of the Baby Showers approach has been published previously [9]. The preparatory phase involved adaptation of standard procedures and tools from research for a programmatic context, site selection, community mobilization, church sensitization, enrollment and training. Once enrolled and trained, sites were activated to begin conducting Baby Shower events. During church service prior to each Baby Shower event, the priest or pastor invited pregnant women and their male partners in the congregation to approach the altar for a prayer session and then to attend the Baby Shower. At the Baby Shower event, potential participants were assessed for eligibility. Inclusion criteria for female participants included a visible pregnancy; women who had previously attended a Baby Shower event in the current pregnancy were excluded. Inclusion criteria for male participants included being a male partner of an eligible pregnant woman. Eligible participants were given information about the Baby Shower event, and verbal consent to participate was ascertained.

The Baby Shower event, typically held after the church service, included group health education, celebratory singing and dancing, gifting of a "mama pack" with safe delivery supplies, and health screening, including weight and blood pressure measurement, as well as HIV and other integrated testing. Baby Receptions were later held as celebrations of birth for new parents and their infants. Trained volunteers called Church Health Assistants, or CHAs, tracked consenting HIV-positive pregnant women and their HIV-exposed infants to support and ensure linkage to PMTCT services including ART at a health facility of their choice.

At Baby Shower events, HIV testing for pregnant women and their male partners was conducted by trained CHAs according to the national HIV testing algorithm using Determine (Abbott Laboratories, IL, US), Uni-Gold (Trinity BioTech, ROI), and Stat-Pak (Inverness Medical-Biostar Inc., DE, US) rapid HIV antibody tests as appropriate [12]. Written consent was obtained for HIV testing. In alignment with Nigerian national guidelines for HIV testing services, young people under the 18 who were married, pregnant or sexually active were considered "mature minors" and were able to give their own consent for HIV testing services.

Pregnant women who tested HIV-positive were asked if they were already on ART. CHAs tracked consenting HIV-positive women not yet on ART to ensure linkage to ART. Where possible, CHAs escorted the women to the health facility, serving as a support system and to observe linkage to ART. Where not possible to escort women (based on distance, timing, or client preference), CHAs tracked women through phone calls and home visits to ask about linkage to ART. Health facility records were not reviewed in this evaluation.

## Data collection and data management

At Baby Shower events, each participant completed a pre-tested semi-structured questionnaire, including socio-demographic information (sex, age, marital status, highest educational attainment, occupation), HIV testing history, and pregnancy history (for women only) (S1 Questionnaires). Questionnaire results and health screening results, including HIV testing, were entered into a de-identified Microsoft Access database with linked participant ID numbers for pregnant women and their male partners. The code linking the de-identified data and participants ID were kept separately in locked storage by study leads to prevent unintended disclosure of protected health information.

Linkage of HIV-positive pregnant women to ART was recorded by CHAs and entered in an Excel database. Women were documented as linked to ART if they were already on ART at the time of testing, were observed to start ART when escorted to the health facility, or verbally reported starting ART during the pregnancy period but after time of testing. For women not linked to ART, the CHA was able to document a reason for lack of ART linkage.

## Data analysis

We reviewed the program data from a cohort of self-identified pregnant women and their male partners who participated in Baby Shower events during the study period and compared these results to program data from PEPFAR-supported PMTCT facilities in the same geographic area for a similar time period.

The Baby Shower and linkage databases were merged in STATA14, and descriptive analyses were conducted with frequencies and proportions for demographic, HIV testing, and pregnancy-related variables. HIV testing coverage, positivity, yield and linkage to ART in community-based Baby Showers events were compared to ANC/PMTCT data from PEPFAR-supported health facilities for approximately the same time period (PEPFAR annual program results for October 2016-September 2017) in the 12 LGAs participating in Baby Showers in

Benue State. PEPFAR data were obtained from PEPFAR publicly available datasets and programmatic reporting [13].

For both congregation-based and facility-based analyses, four key variables are evaluated for pregnant women: HIV testing coverage, HIV positivity, HIV testing yield, and linkage to ART. HIV testing coverage refers to the proportion of pregnant women who have a known HIV status (either newly tested for HIV or previously known to be HIV-positive). HIV positivity among pregnant women refers to the proportion of pregnant women who are HIV-positive, both those who are previously known to be HIV-positive and those who are newly diagnosed. HIV testing yield refers to the proportion of women tested who newly test HIV-positive. Linkage to ART refers to the proportion of HIV-positive pregnant women who are on ART. For this analysis, both women previously on ART and those who newly start ART were considered linked to ART.

## Ethical considerations

Participation in Baby Showers was voluntary. Verbal consent was provided for participation in Baby Shower events, and written consent was obtained prior to HIV testing of participants. HIV testing and subsequent tracing for linkage to care were conducted in alignment with the national HIV testing program guidelines [12]. Ethical approvals from the Nigeria National Health Research Ethics Committee and the Health Research Ethics Committee of the University of Nigeria Teaching Hospital, Enugu, Nigeria were obtained to conduct an analysis of the de-identified program data and publish the findings. The protocol was also reviewed in accordance with the Centers for Disease Control and Prevention (CDC) human research protection procedures and was determined to be research, but CDC investigators did not interact with human subjects or have access to identifiable data or specimens for research purposes.

## Results

### Demographic characteristics

Between July 2016 and October 2017, the enrolled church congregations (n = 80) held 679 Baby Shower events, with a median event size of 30 participants (Interquartile range (IQR) 23 to 46 participants). In total, 16,243 individuals participated in Baby Shower events, including 10,056 pregnant women and 6,187 male partners. Approximately 62% (n = 6,187) of pregnant women attended the Baby Shower with a male partner.

Characteristics of female and male participants in Baby Shower events are shown in Table 1. The majority (80%, n = 8045) of female participants were between 20–34 years of age, and 14% of female participants were less than 20 years of age. No formal education was reported by 18% of female and 6% of male participants. Over 99% of women and men reported they were married. Eighty-eight percent of women and 77% of men identified their occupation as farmer.

### ANC attendance and HIV testing history

Among pregnant women, 55.1% (n = 5,541) reported that they were enrolled in ANC for the current pregnancy. Of 10,055 pregnant women with self-reported gestational age and ANC enrollment status, the proportion of women enrolled in ANC increased based on the trimester of current pregnancy, with 38.1% ANC enrollment among women in the first trimester of pregnancy, 50.4% among women in the second trimester, and 63.7% of women in the third trimester. Nearly one quarter of participants reported that they had never had a HIV test before (22.3%, n = 2,241 female participants and 24.8%, n = 1,532 male participants, respectively).

**Table 1. Characteristics of female and male participants in Baby Shower events.**

|  | Females (n = 10,056) | | Males (n = 6,187) | | Total (n = 16,243) | |
|---|---|---|---|---|---|---|
|  | n | % | n | % | n | % |
| **Age categories, years** | | | | | | |
| <20 | 1442 | 14.3% | 103 | 1.7% | 1545 | 9.5% |
| 20–24 | 3978 | 39.6% | 912 | 14.7% | 4890 | 30.1% |
| 25–29 | 2726 | 27.1% | 1541 | 24.9% | 4267 | 26.3% |
| 30–34 | 1341 | 13.3% | 1406 | 22.7% | 2747 | 16.9% |
| 35–39 | 414 | 4.1% | 978 | 15.8% | 1392 | 8.6% |
| ≥40 | 155 | 1.5% | 1247 | 20.2% | 1402 | 8.6% |
| **Education level** | | | | | | |
| No formal education | 1803 | 17.9% | 351 | 5.7% | 2154 | 13.3% |
| Completed primary school | 2963 | 29.5% | 1126 | 18.2% | 4089 | 25.2% |
| Completed secondary school | 4650 | 46.2% | 3579 | 57.9% | 8229 | 50.6% |
| Attended/completed post-secondary education | 640 | 6.4% | 1131 | 18.3% | 1771 | 10.9% |
| **Occupation** | | | | | | |
| Farmer | 8482 | 84.4% | 4415 | 71.4% | 12897 | 79.4% |
| Trader | 623 | 6.2% | 444 | 7.2% | 1067 | 6.6% |
| Unemployed | 229 | 2.3% | 336 | 5.4% | 565 | 3.5% |
| Civil Servant | 169 | 1.7% | 344 | 5.6% | 513 | 3.2% |
| Other occupation | 260 | 2.6% | 325 | 5.3% | 585 | 3.6% |
| More than one occupation | 293 | 2.9% | 323 | 5.2% | 616 | 3.8% |
| **Marital status** | | | | | | |
| Married | 9946 | 98.9% | 6181 | 99.9% | 16127 | 99.2% |
| Single | 23 | 0.2% | 4 | <0.1% | 27 | 0.2% |
| Separated/Divorced | 20 | 0.2% | 0 | 0.0% | 20 | 0.1% |
| Widowed | 67 | 0.7% | 2 | <0.1% | 69 | 0.4% |

## HIV testing results

Acceptance of HIV testing at the Baby Shower events was nearly universal with 99.9% (n = 16,240) of participants tested for HIV. Overall, 7.2% of female and 4.0% of male participants tested HIV-positive (n = 724 and n = 249 respectively). Based on HIV test results and self-reported HIV testing history from the client questionnaire, 2.8% of females and 2.2% of males were classified as newly diagnosed with HIV (Table 2).

## Linkage of HIV-positive pregnant women to ART

Over 90% of HIV-positive pregnant women (n = 673) were linked to ART, including both women who were previously on ART and those started ART after the Baby Shower. Of 51 HIV-positive pregnant women recorded as not linked to ART, 26 (51.0%) had unknown

**Table 2. HIV testing and results for pregnant women and their male partners participating in Baby Shower events.**

|  | Females (n = 10,056) | Males (n = 6,187) |
|---|---|---|
| Individuals tested, n (%) | 10,055 (>99.9%) | 6,185 (>99.9%) |
| HIV positive, n (%) | 724 (7.2%) | 249 (4.0%) |
| Newly diagnosed, n (%) | 274 (2.9%) | 138 (2.3%) |
| Previously known, n (%) | 450 (4.5%) | 111 (1.8%) |

**Table 3.  PMTCT results from PEFPAR-supported health facilities in the 12 local government areas in Benue State where Baby Showers were conducted, October 2016-September 2017.**

| Indicator | Pregnant women at designated health facilities, N (%) |
|---|---|
| Pregnant women enrolled in ANC | 75676 |
| Pregnant women with HIV status ascertained at first ANC visit | 75469 (99.7%) |
| HIV-positive pregnant women | 6367 (8.4%) |
| Newly diagnosed HIV-positive | 1577 (2.1%) |
| Previously known HIV-positive | 4790 (6.3%) |
| HIV-positive pregnant women on ART* | 6375 (100.1%) |

*The number of HIV-positive pregnant women on ART may be greater than number of HIV-positive pregnant women, as in this case, due to either programmatic data quality issues or because women may be diagnosed and started on ART in different reporting periods for these cross-sectional indicators.

reasons for failed linkage. Among the remaining women, reasons provided for not being linked to ART were unable to reach client (52%, n = 13), not ready because of issues with their male partner (20%, n = 5), client moved (8%, n = 2), and refused ART (8%, n = 2).

## Facility-based ANC/PMTCT data in Benue State

In the same geographic area and time period as Baby Shower events, health facilities reported greater than 99% ascertainment of HIV status among pregnant women at first ANC visit and HIV positivity among pregnant women of 8.4%, with 2.2% of women tested receiving a new HIV-positive diagnosis and all HIV-positive women linked to ART (100.1%) (Table 3).

## Discussion

With training and ongoing implementation support, eighty churches in Benue State carried out Baby Showers events involving over sixteen thousand participants, nearly all of whom received HIV testing in addition to other educational and supportive interventions to improve maternal, child and family health. Nearly one thousand participants were identified as HIV-positive. CHAs provided supportive counseling for those already on ART to encourage ART adherence and supported linkage to ART for those who were newly diagnosed, with over 90% linkage to ART among HIV-positive pregnant women.

The HIV positivity of Baby Shower participants was slightly higher than HIV prevalence in Benue State established in the NAIIS survey (6.3% for females and 3.5% for males age 15–64 years) [3]. A significant proportion of female Baby Showers participants had established risk factors for HIV and/or low PMTCT uptake, including young age, low educational attainment, and not attending ANC, indicating that the Baby Showers included many women and families at risk [14–16]. Pregnant women participating in Baby Showers in Benue State reported lower ANC attendance (55.1% vs. 79.6%) and higher HIV positivity (7.2% vs. 2%) when compared to women enrolled in the Baby Showers trial in Enugu State, Nigeria in 2013–2014 [9]. Thus, Benue State and the selected sites seemed well-suited for this intervention, and these church-based events captured even more missed opportunities for PMTCT intervention than the previous cluster randomized trial that demonstrated the approach's effectiveness.

The high acceptability of HIV testing using the congregation-based approach may be due to the influential role of religious leaders in supporting the Baby Showers as well as the engagement of the CHAs, volunteers who are active members of the church congregation and who

provided counseling and support for women and their partners. Community and peer supporters have been shown to be effective in promoting uptake of PMTCT services [4, 17]. Linkage to ART for HIV-positive pregnant women in the Baby Showers implementation, while lower than in facility-based care in Benue State, was considerably higher than reported linkage from other community-based HIV testing interventions [18, 19]. Providing HIV results confidentially as part of a group, celebratory event is challenging, and participants who are newly diagnosed HIV-positive require ongoing support. Strengthening the role of CHAs in supporting participants' psychosocial needs and linkage and retention on ART is a priority for this approach moving forward.

While male participation in Baby Showers events in Benue State was lower than in the HBI trial (62% vs. 89% male participation, respectively) [20], male participation in facility-based ANC/PMTCT services in sub-Saharan Africa are often reported around or below fifty percent, with significant variation by country and level of programming around male engagement [21–23]. Previous studies demonstrate that male participation in ANC and PMTCT services improve HIV testing uptake, PMTCT retention and infant outcomes, as well as male partner and family health [24–27]. The majority of men who tested HIV-positive in Baby Showers in Benue State were newly diagnosed (138 newly diagnosed HIV-positive male partners out of 249 HIV-positive overall), demonstrating the opportunity of this intervention for finding men and linking them to ART. In a published analysis of the Benue Baby Showers events focused on couples HIV testing results and male participation, the importance of testing male partners of HIV-positive women is noted, and future studies to understand the socio-cultural contexts that enhance male participation in Baby Shower events and HIV testing are recommended [28]. Additionally, index testing of male partners of HIV-positive women, even those who may not attend the events, could be incorporated into this intervention in future.

Extensive data were collected during the Baby Showers implementation in Benue State, with resources dedicated to data collection and management; yet, these data were limited by the biases of self-reported questionnaires, especially for HIV testing history, and contain some data quality gaps. For example, participants may report that they were newly diagnosed with HIV at the Baby Shower event, then later share that they were already on ART during the follow up visits to ensure ART linkage. For this analysis, the information on HIV testing history reported on the Baby Shower questionnaire was used, despite these known limitations. While data from Baby Showers events was of high quality, documentation of subsequent linkage to ART was less complete, and the extent of CHA tracking efforts as well as reasons for lack of linkage were often unclear.

The comparison of Baby Showers results to facility ANC/PMTCT data is imperfect since the congregational sites and health facilities may not have the exact same catchment area and represent slightly different time periods. Neither clients at health facilities or participants in congregational sites are necessarily representative of the general population in the geographic area. Despite these limitations, we report on a large number of pregnant women, and it is worth noting that the positivity and yield for HIV testing are comparable between the facility and community-based HIV testing. The Baby Showers identified many HIV-positive individuals not receiving facility-based services; in this and similar settings with low ANC attendance, community-based approaches to MCH and PMTCT care are particularly important to complement facility services, with opportunities for communities to link individuals to health facilities and also for communities to provide ongoing support for retention in clinical care.

Despite significant financial investments in HIV programs, we were far from reaching the UNAIDS and PEPFAR Start Free-Stay Free-AIDS Free target of less than 20,000 new child HIV infections in 2020 [1]. The health care systems in many low and middle-income countries are not equipped to manage HIV care alone. Community engagement and community-based

interventions are needed to ensure high quality, client-centered care, as well as to share some of the health facility workload in terms of ART adherence support and retention tracking.

The Baby Showers approach uses multicomponent events (not focused on HIV alone) and subsequent follow-up by peer volunteers in Christian church congregations to improve PMTCT uptake, but may be adapted for other settings and health challenges. Baby Showers may incorporate other HIV-related services, such as testing other children in the family and connection with orphans and vulnerable children (OVC) programming, as well as non-HIV related services. The Baby Shower approach exemplifies innovations that leverage social networks and faith-based structures to expand testing and linkage to care for a high burden health condition. Given the large Muslim population in Nigeria, there is proposed work to adapt Baby Showers for the Muslim community and mosques, and this approach could be expanded to other religious and community institutions. Broad maternal, child and sexual health interventions, integrated chronic disease screenings, and evaluation for tuberculosis and other stigmatized conditions could be addressed through similar faith-based and community-oriented approaches.

The Baby Showers approach and similar integrated, multicomponent interventions can improve the efficiency of service provision at the community level. Health screenings, education, linkage to facility care, and client tracking can be done for communicable and non-communicable health conditions, using the same manpower and resources. Such approaches have the potential to improve the efficiency, quality, acceptability, and sustainability of care, especially in resource-limited settings.

## Supporting information

**S1 Dataset. This is the Excel data file for Baby Showers participants and linkage of HIV-positive pregnant women to ART.**
(XLSX)

**S1 Questionnaires. This file contains the questionnaires used in the Baby Shower events, including the registration/biodata form (used for both female and male participants), the male health questionnaire, the female health questionnaire, and the clinical TB and HIV screening form (used for both female and male participants).**
(PDF)

## Acknowledgments

The authors acknowledge the support from the clergy leadership, community leaders at participating sites, staff of the participating health facilities, the pregnant women and their partners. The authors are indebted to the many staff members of Caritas Nigeria and the Center for Translation and Implementation Research (CTAIR) of the University of Nigeria, Nsukka, Enugu who made the Baby Shower events a joyful experience for the participating pregnant women and their partners.

**Disclaimer:** The findings and conclusions in this paper are those of the authors and do not necessarily represent the official position of the funding agencies.

## Author Contributions

**Conceptualization:** Michele Montandon, Timothy Efuntoye, Ijeoma U. Itanyi, Chima A. Onoka, Chukwudi Onwuchekwa, Jerry Gwamna, Echezona E. Ezeanolue.

**Data curation:** Michele Montandon, Ijeoma U. Itanyi, Chukwudi Onwuchekwa.

**Formal analysis:** Michele Montandon, Ijeoma U. Itanyi.

**Project administration:** Timothy Efuntoye, Ijeoma U. Itanyi, Chima A. Onoka, Chukwudi Onwuchekwa, Chibuzor Onyenuobi, Amaka G. Ogidi, John Okpanachi Oko, Echezona E. Ezeanolue.

**Resources:** Michele Montandon, Timothy Efuntoye, Chima A. Onoka, Jerry Gwamna, Mahesh Swaminathan, Echezona E. Ezeanolue.

**Writing – original draft:** Michele Montandon.

**Writing – review & editing:** Michele Montandon, Timothy Efuntoye, Ijeoma U. Itanyi, Chima A. Onoka, Chukwudi Onwuchekwa, Jerry Gwamna, Amee Schwitters, Chibuzor Onyenuobi, Amaka G. Ogidi, Mahesh Swaminathan, John Okpanachi Oko, Gbenga Ijaodola, Deborah Odoh, Echezona E. Ezeanolue.

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
