## [Decision Letter · Decision Letter 0]

14 Jun 2021

PONE-D-20-35814

Improving uptake of prevention of mother-to-child HIV transmission services in Benue State, Nigeria through a faith-based congregational strategy

PLOS ONE

Dear Dr. Montandon,

Thank you for submitting your manuscript to PLOS ONE. After careful consideration, we feel that it has merit but does not fully meet PLOS ONE’s publication criteria as it currently stands. Therefore, we invite you to submit a revised version of the manuscript that addresses the points raised during the review process.

Both reviewers have identified a number of points that should be addressed thoroughly in a revision of your manuscript. Please ensure you carefully work through all of the issues they have highlighted.

We look forward to receiving your revised manuscript.

Kind regards,

Jamie Males

Staff Editor

PLOS ONE

Journal Requirements:

3. In your Methods section, please provide additional information about the participant recruitment method and the demographic details of your participants. Please ensure you have provided sufficient details to replicate the analyses such as:

- a description of any inclusion/exclusion criteria that were applied to participant recruitment

- a statement as to whether your sample can be considered representative of a larger population.

4. In the Methods, please clarify that participants provided oral consent. Please also state in the Methods:

- Why written consent could not be obtained

- Whether the Institutional Review Board (IRB) approved use of oral consent

- How oral consent was documented

For more information, please see our guidelines for human subjects research: https://journals.plos.org/plosone/s/submission-guidelines#loc-human-subjects-research

5. You indicated that you had ethical approval for your study. In your Methods section, please ensure you have also stated whether you obtained consent from parents or guardians of the minors included in the study or whether the research ethics committee or IRB specifically waived the need for their consent.

Reviewers' comments:

Reviewer's Responses to Questions

**Comments to the Author**

1. Is the manuscript technically sound, and do the data support the conclusions?

Reviewer #1: Partly

Reviewer #2: Yes

2. Has the statistical analysis been performed appropriately and rigorously? 

Reviewer #1: N/A

Reviewer #2: Yes

3. Have the authors made all data underlying the findings in their manuscript fully available?

Reviewer #1: Yes

Reviewer #2: Yes

4. Is the manuscript presented in an intelligible fashion and written in standard English?

Reviewer #1: Yes

Reviewer #2: Yes

5. Review Comments to the Author

Reviewer #1: This is a well-written manuscript, describing an effective initiative that was evaluated through a methodologically sound process. It should be published. However, there are some elements that require clarification.

1) P. 5, line 71: Can the authors note whether the baby shower event reflects an existing cultural practice in Nigeria, whether it is an adaptation of such a practice, or whether this is a new event introduced into the community.

2) P 6, line 93ff: Please say a bit more about your selection process of these congregations. How many churches were considered initially? What exclusion criteria were utilized to the this number down to the 80 eventually chosen? How large were these congregations? Were they all Roman Catholic parishes (and if so, it would be helpful to note whether plans are underway to expand this to non-Catholic Christian communities when noting that such plans are underway to expand to Muslim communities).

3) P.7, lines 109-112: Did you assess whether finding out HIV status in a public setting such as this might elicit worry or trigger negative feelings (e.g., shame) in women or men who test positive? Some discussion on considerations regarding this possibility would be helpful if you did indeed consider it.

4) P. 9, lines 150-159: The "Ethical Considerations" sub-section seems misplaced to me as a reader. Shouldn't it be placed ahead of "Data collection and management?"

5) P. 11, lines 188-193: It would interesting to include a specific breakdown of the percentage of women who were newly diagnosed who did not follow up with referral to care. As is, your discussion of the 51 who did not follow up doesn't distinguish between those newly diagnosed and those already aware of their HIV status.

6) P. 13, lines 230-231. The participation rate among men was far lower in your program in Benue State than that of men in the trial. Do you have any data as to why this is so or do you have any assumptions (even if not verified with data) that you could include in a discussion as to possible reasons for this discrepancy?

7) P 14, lines, lines 244-246: You note that the documentation of linkage to care and reasons for unsuccessful linkages was not robust or clear. Can you discuss how you intend to address this in subsequent baby shower programs?

8) THIS IS THE MOST IMPORTANT ELEMENT I WOULD ASK YOU TO ADDRESS: In reading the manuscript, I noted from as early as page 3 (lines 41-43) that comparing results from the baby showers to results from ANC clinics was a bit of an "apple/orange" comparison. I was glad to see the authors acknowledge this themselves and note that the results from the baby shower programs was significantly higher than those from other community based programs (p. 13, lines 226-228). I believe you need to highlight this more clearly and earlier in your manuscript. Clearly lay out that you comparison is indeed looking at cohorts from two very different settings (one clinical and one community-based). Explain why you are making this comparison (I assume you don't have access to the data from similar community initiatives-- if you can access these data, I would include you to include a discussion comparing your program to those in other community settings), and highlight the outcomes from your program (which are impressive as reported) as yielding comparable results as those from a clinical setting. In short, please describe the issue of the differences in setting and how your program yielded outcomes that were nonetheless comparable to those from a clinical program.

Reviewer #2: I find this a very interesting research project which is written up clearly in this article. I think there could be a bit more literature cited and discussed in relation to the unique role of churches and church organizations in responding to HIV&AIDS. I recommend the authors look at my work on this in South Africa and browse the references for other material. It can be cited as follows: Deborah Simpson (2018) “Bringing back hope”: how faith-based responses to HIV and AIDS differ from secular responses, African Journal of AIDS Research, 17:2, 175-182,

DOI: 10.2989/16085906.2018.1478313 and the link to this article is available here: https://doi.org/10.2989/16085906.2018.1478313. Though this is a qualitative study, I believe it will be useful in framing why religious leaders carry so much sway in HIV education, testing, and advocacy. Other than that, it is a convincing and clear article which nicely captures an interesting PEPFAR-funded research project.

6. PLOS authors have the option to publish the peer review history of their article (what does this mean?). If published, this will include your full peer review and any attached files.

Reviewer #1: **Yes: **John B. Blevins

Reviewer #2: **Yes: **Dr. Deborah Simpson

---

## [Author Response · Author response to Decision Letter 0]

23 Jul 2021

Dear Editors/Reviewers,

Thank you for your careful review of our manuscript entitled “Improving uptake of prevention of mother-to-child HIV transmission services in Benue State, Nigeria through a faith-based congregational strategy” (PONE-D-20-35814).

We greatly appreciate your comments. We have revised the manuscript and provided responses to each of the points below.

Journal requirements

We reviewed the PLOS ONE style requirements and have ensured that the manuscript fulfills all criteria. To the best of our knowledge, the reference list is complete and current. We added an additional reference as suggested by Reviewer 2, noted below.

The Baby Shower questionnaires have been added as Supporting Information 2 and referenced in the Methods. This includes the Registration/biodata form (used for both female and male participants), the male health questionnaire, the female health questionnaire, and the clinical TB and HIV screening form (used for both female and male participants).

3. In your Methods section, please provide additional information about the participant recruitment method and the demographic details of your participants. Please ensure you have provided sufficient details to replicate the analyses such as:

- a description of any inclusion/exclusion criteria that were applied to participant recruitment

- a statement as to whether your sample can be considered representative of a larger population.

Thank you for noting the need for additional information on recruitment and eligibility. This was added to the Methods section. The subheading was changed to “Description of Baby Showers implementation” to include recruitment, eligibility, and enrollment.

The demographic details of participants are found in Table 1.

A statement noting that participants in congregational sites are not necessarily representative of a larger population was noted as a limitation in the Discussion section (line 268).

4. In the Methods, please clarify that participants provided oral consent. Please also state in the Methods:

- Why written consent could not be obtained

- Whether the Institutional Review Board (IRB) approved use of oral consent

- How oral consent was documented

For more information, please see our guidelines for human subjects research: https://journals.plos.org/plosone/s/submission-guidelines#loc-human-subjects-research

We have clarified the consent process in the Methods section “Description of Baby Showers implementation.” Oral consent to participate in the Baby Showers event was obtained from all participants, and written consent was obtained prior to HIV testing. Following oral consent, participants were entered in a participant log and assigned a Member ID # that was used on all subsequent documentation (see Supplemental Information 2). Written consent forms were kept in a locked cabinet in a study office in Enugu, Nigeria. This consent process was approved by the Nigeria National Health Research Ethics Committee, the Health Research Ethics Committee of the University of Nigeria Teaching Hospital, and the US Centers for Disease Control. 

5. You indicated that you had ethical approval for your study. In your Methods section, please ensure you have also stated whether you obtained consent from parents or guardians of the minors included in the study or whether the research ethics committee or IRB specifically waived the need for their consent.

We added this sentence in the Methods section to answer this question: “In alignment with Nigerian national guidelines for HIV testing services, young people under the 18 who were married, pregnant or sexually active were considered “mature minors” and were able to give their own consent for HIV testing services.” This process for consent of minor was approved by research ethics committees.

Reviewer 1

1) P. 5, line 71: Can the authors note whether the baby shower event reflects an existing cultural practice in Nigeria, whether it is an adaptation of such a practice, or whether this is a new event introduced into the community.

While baby showers (receptions held in honour of a pregnant woman where she plays pregnancy-related games and receives gifts from friends, usually, items she would need during delivery or immediately after birth) occur commonly in Nigeria, church-organized group baby showers are not typical. Adapting the personal baby shower to a church-based group event that incorporates health screenings and HIV testing is part of the innovation of this approach.

To clarify this point, we altered the description of the Baby Shower intervention in the introduction section (line 72) to state:

“The approach modified the baby shower that commonly occurs in the community (a reception held in honour of a pregnant woman where she plays pregnancy-related games and receives gifts from friends) into a celebratory gathering held at the church, with prayer, singing, dancing, group education, and health screening, including HIV testing for pregnant women and their male partners.”

2) P 6, line 93ff: Please say a bit more about your selection process of these congregations. How many churches were considered initially? What exclusion criteria were utilized to the this number down to the 80 eventually chosen? How large were these congregations? Were they all Roman Catholic parishes (and if so, it would be helpful to note whether plans are underway to expand this to non-Catholic Christian communities when noting that such plans are underway to expand to Muslim communities).

Thanks for these questions. We added to the Methods section that 101 churches were evaluated, and of these, 80 churches were selected as well-suited for Baby Showers implementation. As noted in the methods, the churches were selected based on the capacity/willingness of church (includes adequate space and volunteers), congregational size, and accessibility, including proximity to a health facility. The 21 churches that were evaluated but not selected did not meet one or more of these criteria. The churches were both NKST (translates to “Universal Reformed Christian Church," a Christian Reformed church based in Nigeria) and Catholic, the two main churches in Benue State. The initial RCT in Enugu state also involved both Catholic and non-Catholic churches.

3) P.7, lines 109-112: Did you assess whether finding out HIV status in a public setting such as this might elicit worry or trigger negative feelings (e.g., shame) in women or men who test positive? Some discussion on considerations regarding this possibility would be helpful if you did indeed consider it.

There were challenges around providing HIV results in a private, confidential way during a celebratory, group event. The positive results were typically given toward the end of the event, and the church health assistants (CHAs) followed up with positive clients as noted in the paper to provide additional support and ensure linkage to treatment, since it is often hard to process the results on the first day. We are also planning to publish a manuscript/brief on implementation lessons learned (i.e. taking the Baby Showers from research to practice) that will delve into these issues further. We agree that it is an important point to mention, however, and we edited an existing paragraph in the discussion to include this point (line 241-244):

“Providing HIV results confidentially as part of a group, celebratory event is challenging, and participants who are newly diagnosed HIV-positive in any circumstance and setting require ongoing support. Strengthening the role of CHAs in supporting participants’ psychosocial needs and linkage and retention on ART is a priority for this approach moving forward.”

4) P. 9, lines 150-159: The "Ethical Considerations" sub-section seems misplaced to me as a reader. Shouldn't it be placed ahead of "Data collection and management?"

This is not specified in author instructions, and it appears that published manuscripts from PLOS ONE include the ethics section in different parts of methods. Because the ethical review involved all aspects of methods (including data management and analysis), we included it at the end of the methods section, but are open to changing the order as preferred by the editor.

5) P. 11, lines 188-193: It would interesting to include a specific breakdown of the percentage of women who were newly diagnosed who did not follow up with referral to care. As is, your discussion of the 51 who did not follow up doesn't distinguish between those newly diagnosed and those already aware of their HIV status.

Thank you for this excellent point. We would have liked to do this breakdown, but the distinction between newly diagnosed and those already aware of their HIV status was complicated. For the new vs. known data presented in the paper, we use the information from the Baby Shower questionnaire. However, when participants were followed up to ensure linkage to ART, we often received different information about whether they previously knew their HIV status. It was then difficult to decide which self report to use for ART linkage analysis. We alluded to this in the limitations section, but further detailed this issue with the text (line 260):

“…yet, these data were limited by the biases of self-reported questionnaires, especially for HIV testing history, and contain some data quality gaps. For example, participants may report that they were newly diagnosed with HIV at the Baby Shower event, then later share that they were already on ART during the follow up visits to ensure ART linkage. For this analysis, the information on HIV testing history reported on the Baby Shower questionnaire was used, despite these known limitations.”

6) P. 13, lines 230-231. The participation rate among men was far lower in your program in Benue State than that of men in the trial. Do you have any data as to why this is so or do you have any assumptions (even if not verified with data) that you could include in a discussion as to possible reasons for this discrepancy?

One possibility is that in a more routine implementation setting, without the intensive support of a RCT, there was less active recruitment of male participants. 

In a related publication from this study that focuses specifically on male testing (Gbadamosi et al, PLOS ONE, January 2019), the discussion notes: 

“It is unclear why our findings are inconsistent with those of the HBI trial. The different cultural contexts in which HBI was conducted may offer a plausible explanation. Gender norms that have a strong influence on male partners’ involvement in pregnancy-related events[33,34] may be more pronounced in this setting compared to the southeastern region of Nigeria. Future studies to understand the socio-cultural contexts that enhance male participation in HTS may be beneficial in designing culturally acceptable and scalable partner testing interventions.”

The suggestion for further investigation into context of male participation and how to enhance male engagement has been incorporated into the discussion section (line 254).

7) P 14, lines, lines 244-246: You note that the documentation of linkage to care and reasons for unsuccessful linkages was not robust or clear. Can you discuss how you intend to address this in subsequent baby shower programs?

As noted in line 243, strengthening the role of CHAs in supporting linkage and ongoing retention is a priority for the approach moving forward. While the data from the Baby Shower events was closely reviewed, the subsequent linkage tracking was less standardized and tools were not regularly reviewed for quality. In future programs, the linkage tracking would be reviewed with the same rigor as tools from the events.

8) THIS IS THE MOST IMPORTANT ELEMENT I WOULD ASK YOU TO ADDRESS: In reading the manuscript, I noted from as early as page 3 (lines 41-43) that comparing results from the baby showers to results from ANC clinics was a bit of an "apple/orange" comparison. I was glad to see the authors acknowledge this themselves and note that the results from the baby shower programs was significantly higher than those from other community based programs (p. 13, lines 226-228). I believe you need to highlight this more clearly and earlier in your manuscript. Clearly lay out that you comparison is indeed looking at cohorts from two very different settings (one clinical and one community-based). Explain why you are making this comparison (I assume you don't have access to the data from similar community initiatives-- if you can access these data, I would include you to include a discussion comparing your program to those in other community settings), and highlight the outcomes from your program (which are impressive as reported) as yielding comparable results as those from a clinical setting. In short, please describe the issue of the differences in setting and how your program yielded outcomes that were nonetheless comparable to those from a clinical program.

Thank you for this comment – we have tried to clarify this in the text. The reason for comparison is that facility-based services are the existing standard of care for HIV testing in pregnant women (we added to the Introduction, line 87, that the comparison was reviewed “in order to determine how Baby Showers results in the community setting compare to the standard of care.”) Since there are limited resources for PMTCT programs, we need to explore how to prioritize and/or blend facility and community approaches (this is the subject of last three paragraphs of discussion). By comparing this intervention to the standard facility-based approach, we aimed to show that we can achieve comparable HIV testing yield to facility settings and, importantly, reach women who may be missed by health facilities.

Reviewer 2

I think there could be a bit more literature cited and discussed in relation to the unique role of churches and church organizations in responding to HIV&AIDS. I recommend the authors look at my work on this in South Africa and browse the references for other material. It can be cited as follows: Deborah Simpson (2018) “Bringing back hope”: how faith-based responses to HIV and AIDS differ from secular responses, African Journal of AIDS Research, 17:2, 175-182,

DOI: 10.2989/16085906.2018.1478313 and the link to this article is available here: https://doi.org/10.2989/16085906.2018.1478313. Though this is a qualitative study, I believe it will be useful in framing why religious leaders carry so much sway in HIV education, testing, and advocacy. Other than that, it is a convincing and clear article which nicely captures an interesting PEPFAR-funded research project.

Thank you for this suggestion. We have reviewed the articles you mentioned and referenced the article on “Bringing back hope” in the introduction. We acknowledge that the discussion of the role of faith-based organizations and churches is limited in this paper since it focuses on intervention results; however, we do plan for a more general paper on implementation lessons that will incorporate additional context about working with faith-based and church organizations.

---

## [Decision Letter · Decision Letter 1]

16 Nov 2021

Improving uptake of prevention of mother-to-child HIV transmission services in Benue State, Nigeria through a faith-based congregational strategy

PONE-D-20-35814R1

Dear Dr.  Montandon,

We’re pleased to inform you that your manuscript has been judged scientifically suitable for publication and will be formally accepted for publication once it meets all outstanding technical requirements.

Kind regards,

Professor Kwasi Torpey, MD PhD MPH

Academic Editor

PLOS ONE

Additional Editor Comments (optional):

Comments have been adequately addressed

Reviewers' comments:

Reviewer's Responses to Questions

**Comments to the Author**

1. If the authors have adequately addressed your comments raised in a previous round of review and you feel that this manuscript is now acceptable for publication, you may indicate that here to bypass the “Comments to the Author” section, enter your conflict of interest statement in the “Confidential to Editor” section, and submit your "Accept" recommendation.

Reviewer #2: All comments have been addressed

2. Is the manuscript technically sound, and do the data support the conclusions?

Reviewer #2: (No Response)

3. Has the statistical analysis been performed appropriately and rigorously? 

Reviewer #2: (No Response)

4. Have the authors made all data underlying the findings in their manuscript fully available?

Reviewer #2: (No Response)

5. Is the manuscript presented in an intelligible fashion and written in standard English?

Reviewer #2: (No Response)

6. Review Comments to the Author

Reviewer #2: (No Response)

7. PLOS authors have the option to publish the peer review history of their article (what does this mean?). If published, this will include your full peer review and any attached files.

Reviewer #2: **Yes: **Deborah Simpson

---

## [Editor Report · Acceptance letter]

23 Nov 2021

PONE-D-20-35814R1 

Improving uptake of prevention of mother-to-child HIV transmission services in Benue State, Nigeria through a faith-based congregational strategy 

Dear Dr. Montandon:

I'm pleased to inform you that your manuscript has been deemed suitable for publication in PLOS ONE. Congratulations! Your manuscript is now with our production department. 

Kind regards, 

on behalf of

Professor Kwasi Torpey 

Academic Editor

PLOS ONE